# Robbing the Fed: Directly Obtaining Private Data in Federated Learning with Modified Models

**Liam Fowl**[*]
Department of Mathematics
University of Maryland

**Jonas Geiping**[*]
Department of Computer Science
University of Maryland

**Wojtek Czaja**
Department of Mathematics
University of Maryland

**Micah Goldblum**
Center for Data Science
New York University

**Tom Goldstein**
Department of Computer Science
University of Maryland

## Abstract

Federated learning has quickly gained popularity with its promises of increased user privacy and efficiency. Previous works have shown that federated gradient updates contain information that can be used to approximately recover user data in some situations. These previous attacks on user privacy have been limited in scope and do not scale to gradient updates aggregated over even a handful of data points, leaving some to conclude that data privacy is still intact for realistic training regimes. In this work, we introduce a new threat model based on minimal but malicious modifications of the shared model architecture which enable the server to directly obtain a verbatim copy of user data from gradient updates without solving difficult inverse problems. Even user data aggregated over large batches – where previous methods fail to extract meaningful content – can be reconstructed by these minimally modified models.

## 1 Introduction

Federated learning (Konečný et al., 2015), also known as collaborative learning (Shokri & Shmatikov, 2015), is a mechanism for training machine learning models in a distributed fashion on multiple user devices. In the simplest setting, a central *server* sends out model states to a group of *users*, who compute an update to the model based on their local data. These updates are then returned to the server, aggregated, and used to train the model. Over multiple rounds, this protocol can train a machine learning model, distributed over all users, without exchanging local data – only model updates are exchanged. Two central goals of federated learning are to improve training efficiency by decreasing communication overhead and to side-step issues of user-level privacy and data access rights that have become a focus of public attention in recent years (Veale et al., 2018).

Accordingly, many organizations, ranging from large tech companies (McMahan & Ramage, 2017) to medical institutions with especially strict privacy laws, such as hospitals (Jochems et al., 2016), have utilized federated learning to train machine learning models. However, in practice, data privacy is not guaranteed in general, but is dependent on a large number of interdependent settings and design choices specific to each federated learning system. In this work, we focus on the user perspective of privacy, and we study federated learning systems in which the central server is not able to directly view user data.

The key privacy concern for users is whether model updates reveal too much about the data on which they were calculated. Although Kairouz et al. (2021) discuss that "model updates are more focused on the learning task at hand than is the raw data (i.e. they contain strictly no additional information about the user, and typically significantly less, compared to the raw data)", scenarios can be constructed in which the model updates themselves can be inverted to recover their input user

---

[*]Authors contributed equally. Order chosen alphabetically.

information (Wang et al., 2018; Melis et al., 2018). Simple knowledge of the shared model state and model update can be sufficient for such an attack (Zhu et al., 2019; Geiping et al., 2020). These inversion attacks are particularly fruitful if a user's model update is based on a single data point or only a small batch. Accordingly, a strong defense against these attacks is aggregation. The user only reports model updates aggregated over a significant number of local data points, and data from multiple users can be combined with secure aggregation protocols (Bonawitz et al., 2017) before being passed to the server. This ability to aggregate user updates while maintaining their utility is thought to be the main source of security in federated learning. Averaging raw local data in similar amounts would make it unusable for training, but model updates can be effectively aggregated.

Previous inversion attacks typically focus on a threat model in which the server (server here is a stand-in for any party with root access to the server or its incoming and outgoing communication) is interested in uncovering user information by examining updates, but without modifying the federated learning protocol, a behavior also referred to as *honest-but-curious* or *semi-honest* (Goldreich, 2009). In our case, where the party intending to recover user data *is* the server, this "honest" scenario appears contrived, as the server can modify its behavior to obtain private information. In this work, we are thus interested in explicitly *malicious servers* that may modify the model architecture and model parameters sent to the user.

We focus on scenarios in which an agent obtains data without making suspicious changes to the client code or learning behavior. One scenario which enables this threat model involves recently introduced APIs that allow organizations to train their own models using established federated learning protocols (Cason, 2020). In this environment, a malicious API participant can change their model's architecture and parameters but cannot force unsuspecting edge devices to send user data directly.

We introduce minimal changes to model architectures that enable servers to breach user privacy, even in the face of large aggregations that have been previously deemed secure. These changes induce a structured pattern in the model update, where parts of the update contain information only about a fixed subset of data points. The constituent data points can then be recovered exactly, while evading existing aggregation defenses. For architectures that already contain large linear layers, the attack even works directly, modifying only the parameters of these layers.

## 2  LIMITATIONS OF EXISTING ATTACK STRATEGIES

A range of possible attacks against privacy in federated learning have been proposed in recent literature. In the simplest case of *analytic attacks*, Phong et al. (2017b) were among the first to discuss that the input to a learnable affine function can be directly computed from the gradient of its weights and bias, and additional analysis of this case can be found in Qian & Hansen (2020); Fan et al. (2020), and in Section 3.2. However, analytic recovery of this kind only succeeds for a single data point. For multiple data points, only the average of their inputs can be recovered, leading the attack to fail in most realistic scenarios.

*Recursive attacks* as proposed in Zhu & Blaschko (2021) can extend analytic attacks to models with more than only linear layers - a construction also mentioned in Fan et al. (2020). However, these attacks still recover only the average of inputs in the best case. Improvements in Pan et al. (2020) transform linear layers with ReLU activations into systems of linear equations that allow for a degree of recovery for batched inputs to these linear layers, although preceding convolutional layers still have to be deconvolved by recursion or numerical inversion techniques.

Surprisingly, *optimization-based attacks* turn out to be highly effective in inverting model updates. Wang et al. (2018) propose the direct recovery of input information in a setting where the users' model update is the model parameter gradient averaged over local data. In a supervised learning setting, we define this update by $g$ and the loss function over this model by $\mathcal{L}$ with model parameters $\theta$ and data points $(x, y) \in [0, 1]^n \times \mathbb{R}^m$. The server can then attempt recovery by solving the gradient matching problem of

$$\min_{x \in [0,1]^n} ||\nabla_\theta \mathcal{L}(x, y, \theta) - g||^2 \tag{1}$$

and solve this optimization objective using first-order methods or any nonlinear equation solver. Subsequent work in Zhu et al. (2019); Zhao et al. (2020) and Wainakh et al. (2021) proposes solutions that also handle recovery of targets $y$ and variants of this objective are solved for example in Geiping et al. (2020) with cosine similarity and improved optimization and in Jeon et al. (2021)

with additional generative image priors. Reconstruction of input images can be further boosted by additional regularizers as in Yin et al. (2021) and Qian et al. (2021).

Most attacks in the literature focus on the described *fedSGD* setting (Konečný et al., 2015) in which the users return gradient information to the server, but numerical attacks can also be performed against local updates with multiple local steps Geiping et al. (2020), for example against *fedAVG* (McMahan et al., 2017). In this work, we will discuss both update schemes, noting that gradient aggregation in "time", with multiple local update steps, is not fundamentally more secure than aggregation over multiple data points. Previous attacks also focus significantly on learning scenarios where the user data is comprised of images. This is an advantage to the attacker, given that image data is highly structured, and a multitude of image priors are known and can be employed to improve reconstruction. In contrast, data types with weaker structure, such as tabular data, do not lend themselves to regularization based on strong priors, and we will show that our approach, on the other hand, does not rely on such tricks, and is therefore more data-agnostic.

The central limitation of these attack mechanisms is the degradation of attack success when user data is aggregated over even moderately large batches of data (either by the user themselves or by secure aggregation). State-of-the-art attacks such as Yin et al. (2021) recover only $28\%$ of the user data (given a charitable measure of recovery) on a batch size of $48$ for a ResNet-50 (He et al., 2015) model on ImageNet (ILSVRC2012 (Russakovsky et al., 2015)) with unlikely label collisions. The rate of images that can be successfully recovered drops drastically with increased batch sizes. Even without label collisions, large networks such as a ResNet-32-10 (Zagoruyko & Komodakis, 2016) leak only a few samples for a batch size of 128 in Geiping et al. (2020). These attacks further reconstruct only approximations to the actual user data which can fail to recover parts of the user data or replace it with likely but unrelated information in the case of strong image priors.

Further, although all of the previous works nominally operate under an *honest-but-curious* server model, they do often contain model adaptations on which reconstruction works especially well, such as large vision models with large gradient vectors, models with many features (Wang et al., 2018; Zhu & Blaschko, 2021), special activation functions (Zhu et al., 2019; Zhu & Blaschko, 2021), wide models (Geiping et al., 2020), or models trained with representation learning (Yin et al., 2021; Chen et al., 2020). These may be seen as *malicious* models with architectural choices that breach user privacy. In the same vein, we ask, what is the worst-case (but small) modification that can be applied to a neural network to break privacy?

## 3 MODEL MODIFICATIONS

In this section, we detail an example of a small model modification that has a major effect on user privacy, even allowing for the direct recovery of verbatim user data from model updates.

### 3.1 THREAT MODEL

We define two parties: the server $\mathcal{S}$ and the users $\mathcal{U}$. The server could be a tech company, a third party app using a federated learning framework on a mobile platform, or an organization like a hospital. The server $S$ defines a model architecture and distributes parameters $\theta$ for this architecture to the users, who compute local updates and return them to the server. The server cannot deviate from standard federated learning protocol in ways beyond changes to model architecture (within limits imposed by common ML frameworks) and model parameters. We measure the strength of a malicious modification of the architecture using the number of additional parameters inserted into the model. While it is clear that models with more parameters can leak more information, we will see that clever attacks can have a disproportionate effect on attack success, compared to more benign increases in parameter count, such as when model width is increased.

### 3.2 A SIMPLE EXAMPLE

To motivate the introduction of malicious modifications, we start with the simple case of a fully connected layer. A forward pass on this layer is written as $y = Wx + b$ where $W$ is a weight matrix, $b$ is a bias, and $x$ is the layer's input. As seen in Phong et al. (2017a); Qian & Hansen (2020); Fan et al. (2020), when the parameters of the network are updated according to some objective $\mathcal{L}$, the $i^{th}$

row of the update to $W$:

$$\nabla_{W^i}\mathcal{L} = \frac{\partial \mathcal{L}}{\partial y^i} \cdot \nabla_{W^i} y^i = \frac{\partial \mathcal{L}}{\partial y^i} \cdot x,$$

where we use the shorthand $\mathcal{L} = \mathcal{L}(x; W, b)$. Similarly,

$$\frac{\partial \mathcal{L}}{\partial b^i} = \frac{\partial \mathcal{L}}{\partial y^i}\frac{\partial y^i}{\partial b^i} = \frac{\partial \mathcal{L}}{\partial y^i}.$$

So as long as there exists some index $i$ with $\frac{\partial \mathcal{L}}{\partial b^i} \neq 0$, the single input $x$ is recovered perfectly as:

$$x = \nabla_{W^i}\mathcal{L} \oslash \frac{\partial \mathcal{L}}{\partial b^i} \tag{2}$$

where $\oslash$ denotes entry-wise division.

However, for batched input $x$, all derivatives are summed over the batch dimension $n$ and the same computation can only recover $\sum_{t=1}^n \nabla_{W_l^i}\mathcal{L}_t \oslash \sum_{t=1}^n \frac{\partial \mathcal{L}_t}{\partial b_l^i}$ from each row where $\sum_{t=1}^n \frac{\partial \mathcal{L}_t}{\partial b_l^i} \neq 0$, which is merely proportional to a linear combination of the $x_t$'s. However, data points $x_t$ only appear in the average if $\frac{\partial \mathcal{L}_t}{\partial y_t^i}$ is non-zero, a phenomenon also discussed in Sun et al. (2021). If $\mathcal{L}$ has a sparse gradient, e.g. in a multinomial logistic regression, then this structured gradient weakens the notion of averaging: Let $x$ be a batch of data with unique labels $1, \ldots, n$. In this setting $\frac{\partial \mathcal{L}_t}{\partial y_t^i} = 0$ for all $i \neq t$, so that each row $i$ actually recovers

$$x_t = \sum_{t=1}^n \frac{\partial \mathcal{L}_t}{\partial y_t^i} x_t \oslash \sum_{t=1}^n \frac{\partial \mathcal{L}_t}{\partial y_t^i} = \frac{\partial \mathcal{L}_t}{\partial y_t^i} x_t \oslash \frac{\partial \mathcal{L}_t}{\partial y_t^i}. \tag{3}$$

For a batch of $n$ data points with unique labels, we could thus recover all data points exactly for this multinomial logistic regression. We visualize this in Appendix Fig. 11 for ImageNet data Russakovsky et al. (2015) (image classification, 1000 classes), where we could technically recover up to 1000 unique data points in the optimal case. However, this setup is impractical and suffers from several significant problems:

- **Averaging**: Multiple image reconstruction as described above is only possible in the linear setting, and with a logistic regression loss, a loss that depends on sparse logits is used. Even in this restrictive setting, reconstruction fails as soon as labels are repeated in a user update (which is the default case and outside the control of the server), especially if the accumulation size of a user update is larger than the underlying label space of the data. In this case, the server reconstructs the *average* of repeated classes. In Appendix Fig. 11, we see that in the worst-case scenario where all data points fall into the same class, each piece of user data contributes to the gradient equally, resulting in a mashup reconstruction that leaks little private information.

- **Integration**: As stated above, the naive reconstruction is only guaranteed to work only if the linear (single-layer) model is a standalone model, and not within a larger network. If the naive linear model was placed before another network, like a ResNet-18, then gradient entries for the linear layer contain elements averaged over all labels, as the combined network ostensibly depends on each output of the linear layer.

- **Scalability**: on an industrial scale dataset like ImageNet, the naive logistic regression model would require $> 150M$ parameters to retrieve an image from each label, which is of course far from any practical application.

## 3.3 IMPRINTING USER INFORMATION INTO MODEL UPDATES

Nonetheless, the perfect reconstruction afforded by the linear model remains an attractive feature. To this end, we introduce the *imprint module* class of modifications which overcome the previously described issues, while maintaining the superior reconstruction abilities of an analytic reconstruction as described above. Further, the imprint module can be constructed from a combination of commonly used architectural features with maliciously modified parameters that can create structured gradient entries for large volumes of data.

The imprint module can be constructed with a single linear layer (with bias), together with a ReLU activation. Formally, let $\{x_i\}_{i=1}^n = X \in \mathbb{R}^{n \times m}$ be a batch of size $n$ of user data, then a malicious server can define an imprint module whose forward pass (on a single datapoint, $x$) looks like

$$M(x) = f(W_* x + b_*),$$

where $f$ is a standard ReLU nonlinearity. The crux of the imprint module lies in the construction of $W_* \in \mathbb{R}^{k \times m}$ and $b_* \in \mathbb{R}^k$. We denote the $i^{th}$ row (or channel) of $W_*$ and the $i^{th}$ entry of $b_*$ as $W_*^i$ and $b_*^i$, respectively. We then construct $W_*^{(i)}$ so that

$$\langle W_*^i, x \rangle = h(x),$$

where $h$ is *any* linear function of the data where the server can estimate the distribution of values $\{h(x)\}_{x \sim \mathcal{D}}$ of this function on the user data distribution. For example, if the user data are images, $h$ could be average brightness, in which case $W_*^i$ is simply the row vector with entries identically equal to $\frac{1}{m}$. In order to define the entries of the bias vector, we assume that the server knows some information about the cumulative density function (CDF), assumed to be continuous for the quantity measured by $h$. Note this attack does *not* require the server to know the full distribution of user data, but rather can estimate the distribution of *some* scalar quantity associated with the user data (a much easier task). In Appendix Fig. 5, we see this can be quite easy for a server, and can even be done with a small amount of surrogate data. We also stress that the choice of $h$ here is not important to our method. For the purpose of explanation, we will assume that the quantity measured by $h$ is distributed normally, with $\mu = 0, \sigma = 1$. Then, the biases of the imprint module are determined by

$$b_*^i = -\Phi^{-1}\left(\frac{i}{k}\right) = -c_i,$$

where $\Phi^{-1}$ is the inverse of the standard Gaussian CDF. In plain language, we first measure some quantity, like brightness, with the matrix $W_*$. We duplicate this measurement along the $k$ channels (rows) of $W_*$. In the meantime, we create $k$ "bins" for the data corresponding to intervals of equal mass according to the CDF of $h$. Then, the measurement for a given datapoint will land somewhere in the distribution of $h$.

For example, consider the case when the brightness of some image $x_t$ lands between two values: $c_l \leq h(x_t) \leq c_{l+1}$, and no other image in the same batch has brightness in this range. In this situation, we say that $x_t$ alone activates bin $l$. Then, if the image $x_t$ is passed through the imprint module, we have

$$\left( \nabla_{W_*^l} \mathcal{L} - \nabla_{W_*^{l+1}} \mathcal{L} \right) \oslash \left( \frac{\partial \mathcal{L}}{\partial b_*^l} - \frac{\partial \mathcal{L}}{\partial b_*^{l+1}} \right) = x_t + \sum_{s=1}^p x_{i_s} - \sum_{s=1}^p x_{i_s} = x_t, \qquad (4)$$

where images $\{x_{i_s}\}$ are images from the batch with brightness $> c_l$. That is, the difference in successive rows $l, l+1$ of the gradient entry for $W_*$ correspond to all elements with brightness $c_l \leq h(x) \leq c_{l+1}$ (in this case, assumed to be just $x_t$ - see Appendix B for remark). This is because all of $x_t \cup \{x_{i_s}\}$ activate the non-linearity for layer $l$, since all these images have brightness $\geq c_l$, however, only images $\{x_s\}$ have brightness $\geq c_{l+1}$, so only these images activate the non-linearity for layer $l+1$. An interesting biproduct of this setup is that it would be difficult even for hand inspection of the parameters to reveal the inclusion of this module as the server could easily permute the bins, and add random rows to $W_*$ which do not correspond to actual bins and only contribute to model performance. The gradient does not directly contain user data, so that the leak is also difficult to find by analyzing the gradient data and checking for matches with user data therein.

How successful will this attack be? Recall the parameter $k$ defined in the construction of the imprint module. This corresponds to the number of bins the malicious server can create to reconstruct user data. If a *batch* of data is passed through the imprint module, depending on the batch size $n$ used to calculate the update sent to the server, and number of bins, $k$, the server can expect several bins to activate for *only* one datapoint. And the corresponding entries of the gradient vector can be appropriately combined, and inverted easily. The following result quantifies the user vulnerability in terms of the number of imprint bins, $k$, and the amount of data, $n$, averaged in a given update.

**Proposition 1.** *If the server knows the CDF (assumed to be continuous) of some quantity associated with user data that can be measured with a linear function $h : \mathbb{R}^m \to \mathbb{R}$, then for a batch of size $n$ and a number of imprint bins $k > n > 2$, by using an appropriate combination of linear layer and ReLU activation, the server can expect to exactly recover*

$$\frac{1}{\binom{k+n-1}{k-1}}\left[\sum_{i=1}^{n-2} i \cdot \binom{k}{i} \cdot \left(\sum_{j=1}^{\lfloor \frac{n-i}{2} \rfloor} \binom{k-i}{j}\binom{n-i-j-1}{j-1}\right)\right] + r(n,k)$$

*samples of user data (where the data is in $\mathbb{R}^m$) perfectly. Note: $r(n,k)$ is a correction term (see proof for full expansion).*

*Proof.* See Appendix A.1. □

This can be thought of as a *lower* bound on privacy breaches since, often, identifiable information can be extracted from a mixture of two images. Note that increasing the expected number of perfectly reconstructed images requires increasing the number of imprint bins, which in turn requires increasing the number of channels of the matrix $W_*$ and thus the number of parameters. Thus, to visualize the result above in terms of the server-side hyperparameter, $k$, we plot the expected proportion of data recovered as a function of number of bins in Fig. 1(a). Note that this inversion is analytic, and significantly more efficient and realistic than optimization based methods which often require tens of thousands of update steps.

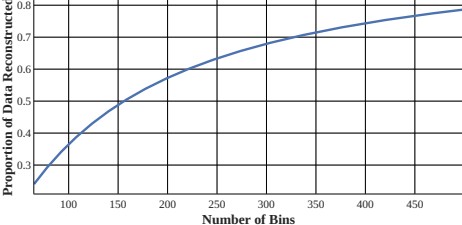 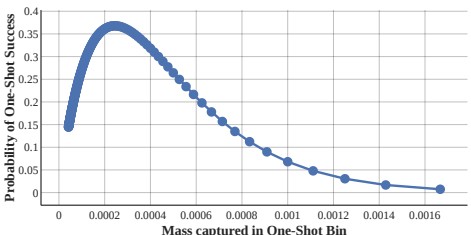

Figure 1: **Left (a)**: Expected proportion of a batch of 64 images *perfectly* recovered as a function of number of bins added via an imprint block in front of a ResNet-18 on ImageNet. With only 156 bins, an attacker can expect to recover over 50% of a batch of user images perfectly. **Right (b)**: Probability of a successful "one-shot" attack on a batch of 4096 images as a function of mass captured in the one-shot bin. An attacker can optimize their bin size given an expected batch size.

The imprint module as described can be inserted in any position in any neural network which receives a non-zero gradient signal. To recover the input feature dimension of the module, a second linear layer can be appended, or – if no additional parameters are of interest – the sum of the outputs of the imprint module can be added to the next layer in the network. The binning behavior of the imprint module is independent of the structure of succeeding layers as long as any gradient signal is propagated. This flexibility allows for wide trade-offs between inconspicuousness and effectiveness of the imprint module. The module can be placed in later stages of model whereas an early linear layer might be suspicious to observers (now that they have seen this trick), but depending on the

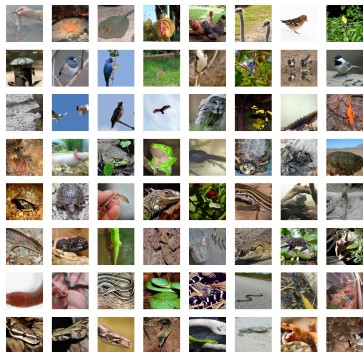 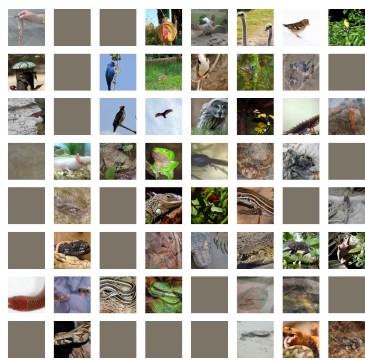

Figure 2: **Left:** Ground truth batch of 64 user images. **Right:** Analytic reconstruction for an imprint model with 128 bins in front of a ResNet-18. Gray reconstructions denote bins in which no data point falls.)

data modality, early linear layers can be a feature of an architecture anyway, in which case there is even no model modification necessary, only parameter changes. The parameter alterations necessary to trigger this vulnerability can furthermore be hidden from inspection until use. The layer can lay "dormant", functioning and training as a normal linear layer initialized with random activations, as long as the server desires. At any point, a party with access to the server can send out parameter updates containing $W^*, b^*$ and trigger the attack.

## 4 EXPERIMENTS

In the following section, we provide empirical examples of imprint modules. In all experiments we evaluate ImageNet examples to relate to previous work (Yin et al., 2021), but stress that the approach is entirely data agnostic. Given an aggregated gradient update, we always reconstruct as discussed in Section 3. When, in the case of imprecision due to noise, more candidate data points are extracted than the expected batch size, only the candidates with highest gradient mean per row in $W^*$ are selected and the rest discarded. For analysis, we then use ground-truth information to order all data points in their original order (as much as possible) and measure PSNR scores as well as Image Identifiability Precision (IIP) scores as described in Yin et al. (2021). For IIP we search for nearest-neighbors in pixel space to evaluate a model-independent distance - a more strict metric compared to IIP as used in Yin et al. (2021). All computations run in single floating point precision.

### 4.1 FULL BATCH RECOVERY

We begin with a straightforward and realistic case as a selling point for our method - user gradient updates aggregated over a batch of 64 ImageNet images. We modify a ResNet-18 to include an imprint module with 128 bins in front. This relatively vanilla setup presents is a major stumbling block for optimization based techniques. For example, for a much smaller batch size of 8, the prior art in optimization based batched reconstruction achieves only 12.93 average PSNR (Yin et al., 2021). We do of course operate in different threat models (although Yin et al. (2021) also uses non-obvious parameter modifications). However, our imprint method is successfully able to recover almost perfect reconstructions of a majority of user data (see Fig. 2), and achieves an average PSNR of 75.75. We further stress that a batch size of 64 is by no means a limitation of the method. If bins, and hence additional rows in $W^*$ are added proportionally to the expected batch size, then recovery of significant proportions (see Fig. 1) of batches of arbitrary size – albeit with the incurred cost in additional parameters for each row – is possible. Note that concurrent work, Boenisch et al. (2021) also proposes a method to recover user data from large-batch updates by malicious modifications - operating in the same setting/threat model as our work . However, in a direct comparison, we find that our imprint module presents a much more significant threat to privacy compared to Boenisch et al. (2021) - see Appendix Fig. 21.

### 4.2 PRIVACY BREACHES IN INDUSTRIAL-SIZED BATCHES – ONE-SHOT ATTACKS

A breach in privacy can occur if even a *single* piece of user data is compromised. Unfortunately, there is a threatening modification of the imprint module for exactly such an attack. If a server has access to enough users, then it becomes feasible for the server to start *fishing* for private data among all updates, and attempt to recover a single data point from each incoming batch of data. Attacks of this nature require only as many additional parameters as twice the size of a single piece of targeted user data, as only *two* bins are needed. For this, $k$ bins are constructed initially, and then all bins are "fused" to create 2 final bins: a one-shot bin containing mass $n/k$, and the other containing the remaining mass $(n+1)/k$. For perspective, for a ResNet-18 on ImageNet, this would require only an additional 1% of parameters - this is tiny compared to potentially massive increases in the number of parameters when no bins were fused, which could raise suspicion under inspection. Further, based on Proposition 1, it is always possible to select an optimal bin size, so that a data point is leaked on average once every four batches of incoming data, *no matter how large the batch size*. We demonstrate this statistical property by recovering a single image from an aggregated batch of $2^{14} = 16,384$ ImageNet images (see Fig. 3). Even though the gradient updates are averaged over a vast number of data points, there can be no perfect privacy, and one data point is leaked in its entirety by only a minor model modification.

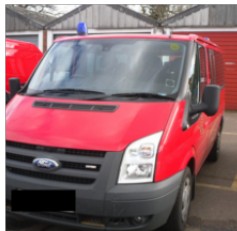 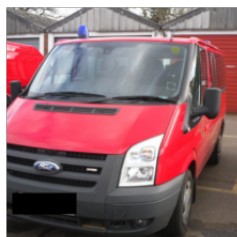

Figure 3: **Left (a)**: A true user image from class "minibus". **Right (b)**: The reconstructed image from class "minibus" captured via our one-shot attack from averages aggregated over **16,384 data-points**. The PSNR is 161.36, i.e. a verbatim copy at machine precision. This user could potentially be identified via their recovered license plate, which was blanked out (by us!) to preserve privacy.

### 4.3 VARIANTS

**Flexible placement** The imprint module does not depend on its placement within a given model. No matter its position in a network, the incoming input features will be leaked to the server. Furthermore, the server can also change the parameters of preceding layers to represent (near)-identity mappings, allowing for the recovery of raw input data from inconspicuous positions deep in a network. For a convolutional network, we show off this variant in Fig. 14 for a ResNet-18. Here, the model parameters are manipulated to contain identity maps up to the location of the imprint module, while the downsampling operations remain. Even these later layers leak enough information that their inputs can be upsampled to breach privacy of the inputs. We remark that we show a simplified version of this attack here where the first three channels in each layer act as an identity, and all other channels are zero, but the model can also be modified to provide off-set pixels in all other channels, effectively increasing the spatial resolution in any layer proportionally with the number of channels.

**Multiple local updates:** In several federated learning protocols, such as fedAVG, users take several local update steps on data before sending model updates to the server Konečný et al. (2015). The imprint module is threatening when a large amount of user data is used for a *single* update step. In this case, the only variable that matters is amount of total data used in the update. That is to say, 10 users sending updates on 100 datapoints each is equivalent (recovery-wise) to a single user sending an update calculated on 1000 datapoints. However, when multiple steps are taken, inverting gradients from pairwise differences (as in Eq. (4)) becomes more difficult, as entries in $W^*$ shift with local updates. However, an imprint variant that produces sparse gradients per data point is a threat to such federated averaging. Defining a forward pass in this new variant as $M'(x)$ as $M'(x) = g(W_* x + b_*)$ where the non-linearity $g$ is a thresholding function:

$$g(t) = \begin{cases} 0 & t \leq 0 \\ t & 0 \leq t \leq 1 \\ 1 & 1 \leq t \end{cases}$$

Note this non-linearity can be simply constructed with a combination of two ReLUs, or with an implementation of a Hardtanh. We now define $W_*^i$ as: $\langle W_*^i, x \rangle = \frac{h(x)}{\delta_i}$ where $h$ is the a linear function of the data as described before, and the biases are defined as:

$$b_*^i = -\frac{c_i}{\delta_i} \quad \text{where} \quad \delta_i = \Phi^{-1}(\frac{i+1}{k}) - \Phi^{-1}(\frac{i}{k}).$$

This setup creates sparse bins where inversion is directly possible (without taking pairwise differences) in gradient entries, at the cost of an additional activation layer. Analyzing a local update step with $W_*^{i,j}$ as the $i^{th}$ row of $W_*$ at update step $j$, reveals that

$$W_*^{i,j} = W_*^{i,j-1} - \alpha \frac{\partial \mathcal{L}}{\partial a^{i,j}} x_j$$

where $x_j$ denotes the data from the previous batch that activated bin $i$, and $a^{i,j}$ denotes the $i^{th}$ activation at step $j$, and $\alpha$ is the local learning rate. Note that if either a linear layer, or a convolutional layer follows this imprint module, then $\frac{\partial \mathcal{L}}{\partial a^{i,j}}$ does not depend on the scale of $W_*^{i,j}$. Therefore, a

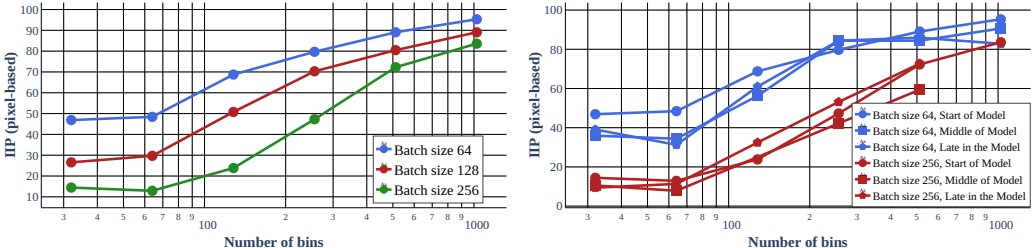

Figure 4: **Left:** Identification Success vs bin size. **Right:** Identification Success (via IIP score) vs. bin size and position in a ResNet-18 model. We find that the attack is stable over a range of batch sizes and positions in a model.

simple way to increase the effectiveness of the new imprint module, $M'$ in the fedAVG case is to scale the linear function associated to the rows of $W_*$ - i.e. $h'(x) = c_0 \cdot h(x)$. This "flattens" the distribution of values, and increases the relative size of $b_*^{i,j}$ compared to the gradient update $\frac{\partial \mathcal{L}}{\partial a^{i,j}}$, which prevents the bins from shifting too significantly during local updates. As bin shift goes to $0$, we recover the situation where the only variable that matters is the total data used in an update.

With this modification, reconstruction quality remains similar to the fedSGD setting. For example, splitting the batch of 64 ImageNet images up into 8 local updates with learning rate $\tau = 1e - 4$ yields an IIP score of $70.31\%$, due to minor image duplications where images hit multiple shifted bins, which we visualize in Appendix Fig. 19.

**Other data modalities:** Yet another advantage to our imprint module over existing optimization based gradient inversion techniques is the flexibility in data domain. Other techniques have demonstrated some success in the image domain by leveraging strong regularizers including total variation (TV), image registration, matching batch norm statistics, and DeepInversion priors (Geiping et al., 2020; Yin et al., 2021). Such strong regularizers do not always exist in other domains of interest, such as text or tabular data. The discussed imprint module, however, is data-agnostic, and while we focus our experiments on the image domain, nowhere do we use any assumptions unique to vision. In fact, linear layers often appear in language models, and tabular data models - cases in which the attacker only needs to modify parameters of an existing model to breach user privacy (Vaswani et al., 2017; Somepalli et al., 2021) without architecture modifications.

## 5 POTENTIAL DEFENSE AND MITIGATION STRATEGIES

If aggregation is the only source of security in a FL system, then the proposed attack breaks it, uncovering samples of private data from arbitrarily large batches, especially via the One-shot mechanism. In light of this attack, the effectiveness of secure aggregation is reduced to only *secure shuffling* (Kairouz et al., 2021): When private data is uncovered via the imprint module, based on data that has been securely aggregated, then the data is breached, but is not directly connected to any specific user (aside from possible revealing information in the data itself). A mitigation strategy for users that does not require coordination (or consent) of a central server is to employ local differential privacy (Dwork & Roth, 2013). Adding sufficient gradient noise can be a defense against this attack as the division in Eq. (2) leads to potentially unbounded errors in the scale of the data. Yet, in practice, privacy is often still broken even if the correct scale cannot be determined, so that the amount of noise that has to be added is large. In Appendix Fig. 20, even with $\sigma = 0.01$, private data is visibly leaked. Additional discussion on defenses can be found in Appendix A.7

## 6 CONCLUSIONS

Federated learning offers a promising avenue for training models in a distributed fashion. However, the use of federated learning, even with large scale averaging, does not guarantee user privacy. Using common and inconspicuous machine learning modules, a malicious server can breach user privacy by sending minimally modified models and parameters in a federated setup. We hope that constructing these examples clarifies current limitations and informs discussions on upcoming applications, especially concerning API design.

## ETHICS STATEMENT

In this work, we uncover an attack on federated learning that has the potential to compromise user privacy. While this method has the potential to be used for malicious purposes, the fundamental purpose of this research is to inform the community about the state of privacy in federated learning, and to help users and technical experts better understand the limitations of federated learning for the purpose of preserving user privacy.

## REPRODUCIBILITY STATEMENT

We provide additional technical details and the proof Proposition 1 in the appendix. Further, we provide open-source implementations of all attacks investigated in this work. The repository at `https://github.com/lhfowl/robbing_the_fed` shows a minimalistic implementation of the principles of this attack and the repository at `https://github.com/JonasGeiping/breaching` embeds the attack in a larger framework of privacy attacks against federated learning. The experiments in this work require no GPU resources and can be cheaply evaluated on almost any machine with sufficient RAM.

## ACKNOWLEDGEMENTS

This work was supported by DARPA GARD, the Office and Naval Research, and the National Science Foundation Division of Mathematical Sciences. Addition support was provided by the Sloan Foundation, JP Morgan Chase, and Capital One Bank.

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

# A  APPENDIX

## A.1  PROOF OF PROPOSITION 1

**Proposition.** *If the server knows the CDF (assumed to be continuous) of some quantity associated with user data that can be measured with a linear function $h : \mathbb{R}^m \to \mathbb{R}$, then for a batch size of $n$, and a number of imprint bins $k > n > 2$, by using an appropriate combination of linear layer and ReLU activation, the server can expect to recover*

$$\frac{1}{\binom{k+n-1}{k-1}}\left[\sum_{i=1}^{n-2} i \cdot \binom{k}{i} \cdot \left(\sum_{j=1}^{\lfloor \frac{n-i}{2}\rfloor} \binom{k-i}{j}\binom{n-i-j-1}{j-1}\right)\right] + r(n,k)$$

*amount of user data (where the data is in $\mathbb{R}^m$) perfectly. Note: $r(n,k)$ is a correction term (see proof for expression).*

*Proof.* By construction of the imprint module, given a random sample (batch) $X_1, \ldots, X_n$ (iid) the server perfectly recovers data whenever an imprint bin has exactly 1 element of the batch. Because we know the CDFs, we can create partitions of equal mass corresponding to imprint bins $\{b_j\} = \{[a_j, b_j]\}$ where $P(X_i \in [a_j, b_j]) = 1/k \; \forall i, j$.

We can then phrase the problem of expected number of perfectly recovered samples as a modified "stars and bars" problem. For a given batch of data, to calculate the amount of data recovered, we first calculate:

$$\sum_{i=1}^{n} i \cdot \binom{k}{i} \cdot N_i$$

Where $\binom{k}{i}$ is the number of ways to select the $i$ bins that have exactly 1 element, and $N_i$ is the number of orientations of the remaining data into the remaining bins so that no bin has exactly 1 element. Note that we can do this because the bins all have equal mass, and thus we can factor out the (uniform) probability of any configuration from the sum.

Simply put, we first take the configuration where there is only 1 bin with exactly 1 element, and weight it by 1, then we take the number of configurations with 2 bins with exactly 1 element, and weight it by 2, and so on.

In order to calculate $N_i$, we notice that in our construction, once the $i$ bins with exactly 1 element are chosen, every other bin has either 0 or $\geq 2$ elements. We focus on the bins that have $\geq 2$ elements. By a simple "reverse" pigeon hole argument, we can now have at most $\lfloor \frac{n-i}{2}\rfloor$ of the remaining bins containing any elements, as otherwise, one bin would be guaranteed to contain exactly 1 element.

So we further select any $1 \leq j \leq \lfloor \frac{n-i}{2}\rfloor$ number of the remaining $k - i$ bins all to contain at least 2 elements. Formally, this is equivalent to calculating the number of orientations of integers $\{x_l\}_{l=1}^{j}$

so that

$$x_1 + \cdots + x_j = n - i$$

constrained with $x_l \geq 2 \; \forall l$

Now, we make a change of variables to instead calculate the number of orientations of integers

$$\{p_l\}_{l=1}^{j}$$

so that

$$p_1 + \cdots + p_j = n - i - 2j$$

constrained with $p_l \geq 0 \; \forall l$. Now we just have a "stars and bars" problem with $k' = j$ bars and $n' = n - i - 2j$ stars. This reduces to:

$$\binom{n-i-j-1}{j-1}$$

orientations for the remaining bins with exactly $j$ elements. Once these $i$ bins with 1 element, and $j$ bins with $\geq 2$ elements are chosen, all the other bins are required to have 0 elements.

| Linear Function | Assumed Distribution | MSE | PSNR | IIP-Pixel |
|:---:|:---:|:---:|:---:|:---:|
| Mean | Normal | 0.0183 | 75.75 | 65.62% |
| Mean | Laplacian | 0.0174 | 79.63 | 71.88% |
| Cosine | Laplacian | 0.0167 | 99.82 | 79.69% |
| Random | Normal | 0.0203 | 91.29 | 75.00% |

Table 1: Ablation study linear functions and distributions.

So adding these parts together, we have the expected amount data the server can expect to reconstruct perfectly becomes:

$$\frac{1}{\binom{k+n-1}{k-1}} \left[ \sum_{i=1}^{n-2} i \cdot \binom{k}{i} \cdot \left( \sum_{j=1}^{\lfloor \frac{n-i}{2} \rfloor} \binom{k-i}{j} \binom{n-i-j-1}{j-1} \right) \right] + \overbrace{\frac{n}{\binom{k+n-1}{k-1}} \binom{k}{n} - \frac{n}{k}}^{r(n,k)}$$

We call the last two "residual" terms $r(n,k)$. The first of these terms corresponds to the term in the expectation where all elements of the batch end up in separate bins, and the second term is the expected number of elements that land in the tail of the CDF not covered in any bin. □

## A.2 Other Choices of Linear Functions and Distributions

In previous experiments we have restricted our investigations to the linear function $h : \mathbb{R}^m \to \mathbb{R}$ that measures average brightness, i.e. $h(x) = \frac{1}{m} \sum_{i=1}^{m} x_i$ which we approximate to be normally distributed. Given that ImageNet (and this also applies to most image datasets) is pre-processed by normalization by color in each channel, and that the number of pixels is large and they are not perfectly correlated, this is a reasonable approximation based on the central limit theorem that could similarly apply to other data modalities as well. For analysis, we visualize the closeness of this approximation based on an evaluation over the full ImageNet validation set in Fig. 6.

We verify that the actual ground truth distribution can be approximated by a normal distribution, but we also see that the approximation is imperfect. The attack works well in Fig. 2 even with this discrepancy, however it could be further improved if the attacker has more accurate about the CDF. Image brightness is better described by a Laplacian distribution Ruderman (1994); Huang & Mumford (1999). Replacing the normal distribution by a Laplacian distribution with scale $1/\sqrt{2}$ does improve the accuracy slightly. This distribution can be further stabilized by considering higher frequencies compared to the mean, e.g. via DCT coefficients Lam & Goodman (2000); Huang & Mumford (1999). We accordingly also visualize this distribution for e.g. the 32nd DCT coefficient in Fig. 6 and use this cosine wave for the imprint module (with scaling factor $\frac{4}{m} f$. This leads to the strongest attack against image data, but of course utilizes attacker knowledge that the users train on natural images.

On the flip side, the estimation can also be improved by replacing the linear function $h$ with a Gaussian random vector of independent draws from $\mathcal{N}(0, \frac{1}{\sqrt{m}})$. The resulting distribution (4th figure in Fig. 6) approximates a normal distribution much better. While not as optimal as the Laplacian distribution for higher frequencies, this variant is applicable for other data modalities if the data has bounded variance. Visualizations of the reconstruction with other linear functions can be found in Fig. 7.

Finally, even with a small amount of data, the server could estimate the density of the quantity of interest. Visually, we plot the the estimated density as for several amounts of ImageNet data used to estimate the brightness distribution. We find that even with 0.1% of the data used, the server could obtain a close approximation to the distribution of interest (see Fig. 5).

## A.3 Comparison to Honest Servers and Optimization-based Attacks

We argue that the proposed attack operating in our threat model is significantly more threatening than existing optimization-based attacks in the honest-but-curious server model. To illustrate this point and show by example that the change is threat model which amounts to only a minor architectural

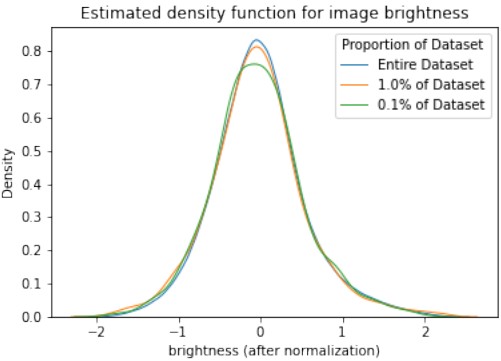

Figure 5: Density of image brightness estimated from access to different amounts of data from the ImageNet dataset.

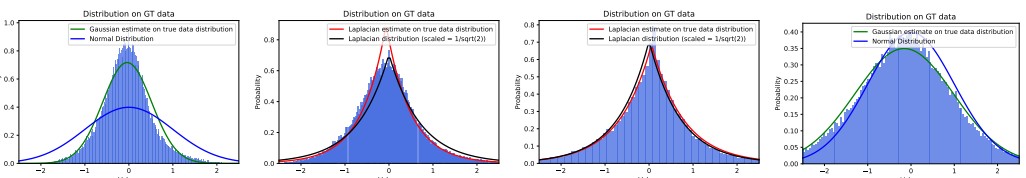

Figure 6: Distributions on the ImageNet validation set for several linear query functions. From left to right: Mean compared to normal distribution, mean compared to Laplacian distribution, 32nd DCT coeffcient compared to Laplacian distribution, random normal vector compared to normal distribution. In each plot the approximate distribution used by the attacker is visualized in blue/black and the ground-truth (GT) distribution in green/red.

change in the neural network leads to a massive difference in reconstruction, we run the attack of Geiping et al. (2020) in the scenario of Fig. 2. The results can be found in Fig. 8. The attack leads to an IIP score of $6.25\%$ when measuring in pixel space, $6.25\%$ when measuring in LPIPS (Zhang et al., 2018) and, and $20.31\%$ when measuring the cosine distances in feature space of this model (the metric of Yin et al. (2021)).

As an additional ablation, we also investigate whether the optimization-based attacks can find the optimal solution in the malicious server threat model that we consider. However, the right side of Fig. 8 shows that at least conventional optimization-based methods have trouble finding the vulnerability introduced by the imprint module. The vulnerability that the attack solves analytically might be hard to exploit by first-order optimization or require specifically tuned optimization schemes to succeed. This also shows that defenses that attempt to detect privacy breaches by evaluating a range of optimization-based attacks would not have triggered an alarm for this attack.

### A.4 OTHER DATA MODALITIES - TEXT

As discussed in the main body, the attack is entirely data-agnostic and could be launched against any kind of input data, e.g. not only image data but also tabular features or text. In principle the input data could be comprised of random signals - these can still be binned and separated by the proposed attack. We verify this property by launching the same attack against text data.

We investigate the transformer architecture discussed in Wang et al. (2021) specifically for language tasks in federated learning scenarios and insert the imprint module as malicious block right after the word embeddings. This is strictly a maliciously modified architecture - normal feedforward blocks in a transformer would not span across the entire length of the sequence. We initialize the linear function as a Gaussian random vector and make no modifications to the attack hyperparameters. We recover the input tokens ids from a direct lookup of their closest match in the word embedding layer (Zhu et al., 2019). To evaluate the success of the attack we generate sample batches of sentences

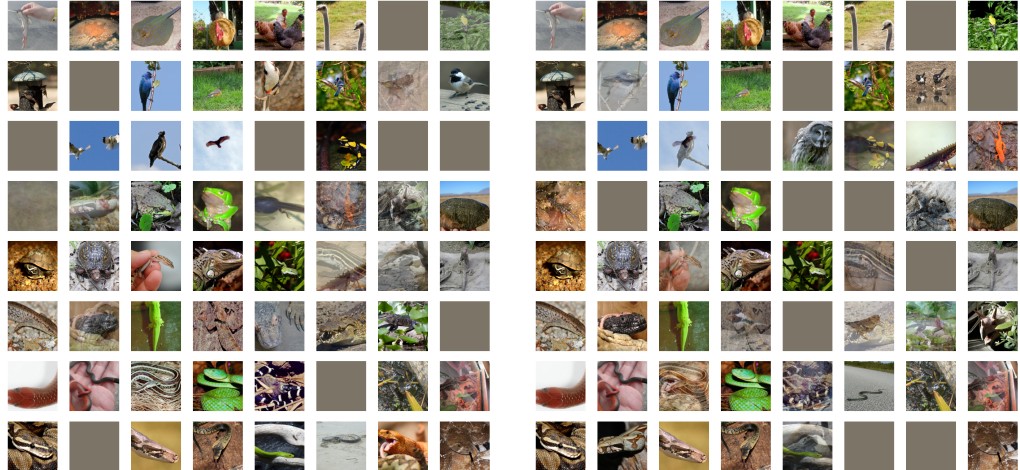

Figure 7: Analytic reconstruction for an imprint model with 128 bins in front of a ResNet-18. **Left:** The linear function is the 32nd DCT coefficient and bins are based on a Laplacian distribution. **Right:** Linear function is a Gaussian random vector and bins are based on a normal distribution. Gray reconstructions denote bins in which no data point falls.)

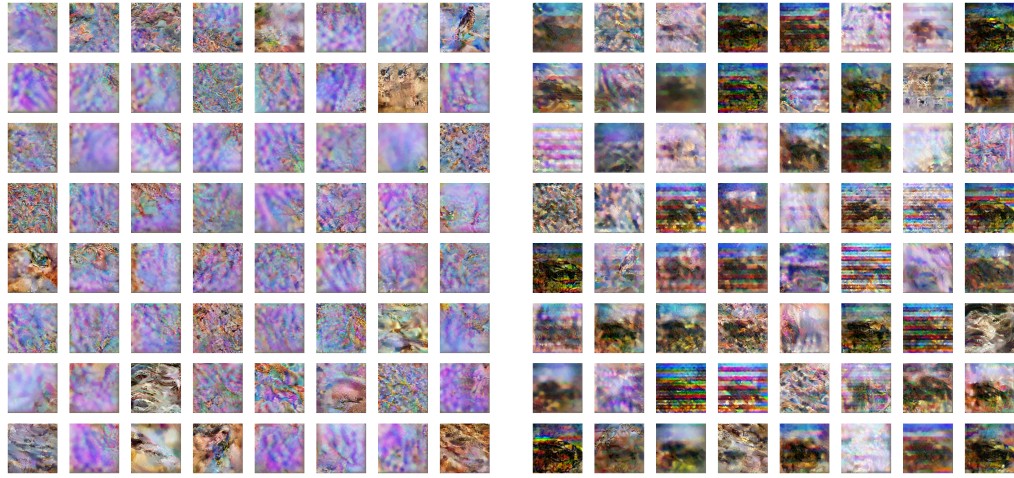

Figure 8: Optimization-based attack of Geiping et al. (2020) for a ResNet-18 and a batch size of 64, the setting of Fig. 2. **Left:** An honest server model. **Right:** Gradient inversion attack applied to a module that contains the imprint module.

from the wikitext dataset (Merity et al., 2016) with a batch size of 128 and a sequence length of 32, tokenized via the GPT-2 tokenizer (Radford et al., 2019). Instantiating the attack with 512 bins immediately reveals 110 out of 128 sentences perfectly, leading to an overall reconstruction accuracy of 86.33% which is also a BLEU score of 88% and ROUGE-L of 87%. We refer to our open source implementation for further details.

We show examples of recovered data below, but note that printing the recovered text is not that insightful. The attack perfectly recovers a subset of sentences of user text and cannot recover some other sentences entirely, in full analogy to the results for images in e.g. Fig. 7:

Recovered wikitext data:
```
The Tower Building of the Little Rock Arsenal, also known as
U.S. Arsenal Building, is a building located in MacArthur Park
in downtown Little Rock, Arkansas .  Built in 1840, it was
part of Little Rock's first military installation.  Since its
decommissioning, The Tower Building has housed two museums.  It
was home to the Arkansas Museum of Natural History and Antiquities
from 1942 to 1997 and the MacArthur Museum of Arkansas Military
History since 2001.  It has also been the headquarters of the
Little Rock Æsthetic Club since 1894.  The building receives its
name from its distinct octagonal tower.  Besides being the last
'''''''''''''''''''''''''''''' ''''''''''''''''''''''''''''''''
```

## A.5 TECHNICAL DETAILS

All experiments were implemented in PyTorch (Paszke et al., 2017) and were run on several laptop and machine CPUs, as the reconstruction itself requires only a few tensor operations. Especially, compared to optimization-based reconstruction techniques, this makes the approach significantly faster and significantly more portable. For visualization purposes and to measure accurate PSNR and IIP scores all images (which are recovered in the order given by the chosen function $h$) are matched with possible correspondences in the ground truth batch. The matching is found based on LPIPS feature similarities scores (Zhang et al., 2018) which are matched using a linear sum assignment solver. No labels are recovered using the proposed approach which is entirely label-agnostic, but labels could be assigned a-posteriori using model predictions of the reconstructed data if required. For experiments where the imprint module is placed deeper into a network, the network (which is here a ResNet) is linearized by resetting batch normalization parameters and buffers to the identity map, setting all residual paths to zero and initializing the first convolution and the shortcut convolutions to identity maps. The nonlinearities can be bypassed by bias shifting as in Goldblum et al. (2020).

PSNR scores are computed as average PSNR where we first compute PSNR scores per image and then average. This procedure is standard in computer vision, but does bias the score toward success-ful reconstructions, as the minimal PSNR score is bounded at 0, but its potential upside unbounded. For the image identifiability precision (IIP) score of Yin et al. (2021) we implement the score as proposed therein, but measure nearest neighbors not in the model feature space (which we consider biased, given that the model parameters are already used for reconstruction), but directly in image pixel space, where we check whether the given reconstruction is indeed closer to its true counterpart in euclidean distance than any other image from this class in the validation set.

## A.6 ADDITIONAL IMAGES

This section contains additional image examples, such as using CIFAR-10 in Fig. 12 and Fig. 13 where an attack with the imprint module on this dataset shows that almost all of the user data is perfectly recovered. Furthermore, example panels of imprint modules inserted in later ResNet layers are visualized as well as the results of the sparse variant that is used to attack a federated averaging scheme.

## A.7 DEFENSE DISCUSSION
An algorithmic defense against the proposed attack would be to validate the incoming model pa-rameters on the user side. There, the attack with multiple bins requires repeated computations of

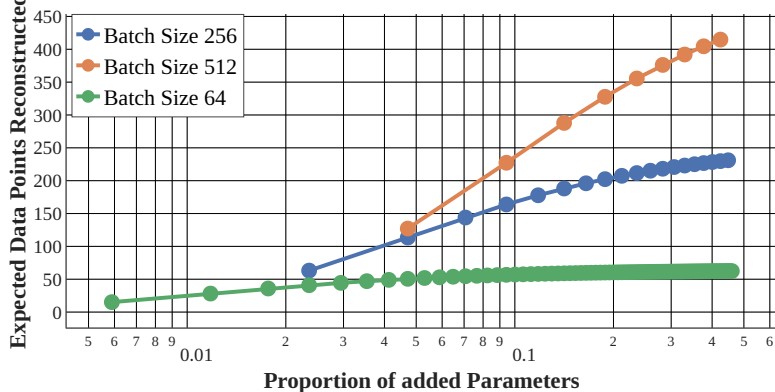

Figure 9: Expected number of data points recovered for several batch sizes and increased bins and corresponding parameter increase. Model: ResNet50 with ImageNet, targeting the input to the 3rd residual block.

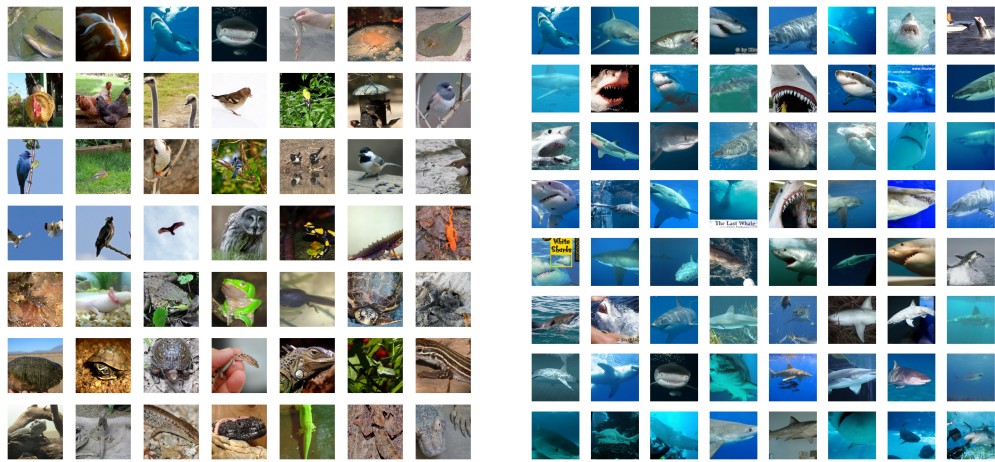

Figure 10: **Left:** Raw data for the 64 ImageNet images with separate classes. **Right:** Raw data for the 64 images from the white shark class.

the same quantity. At first, this pattern could be detected by rank analysis of all linear layers (which would return a rank of 1 for the linear layer of the imprint module described above). Yet, the attacker can easily randomize a few entries of the linear layer to increase its rank without significantly weakening the attack, or introduce additional rows that compute normal deep features, so that we do not believe a defender can win by model analysis under the given threat model.

## A.8 COMPARISON TO CONCURRENT WORK

We include a comparison to the concurrent attack of Boenisch et al. (2021) operating in the same threat model in Fig. 21. The threat model is characterized purely as a parameter modification therein that only applies to models with input-sized linear layers followed by ReLU activations. In the same vein the attack discussed in this work does not require malicious architecture modifications if these vulnerable layers are already present.

## B REMARK ON RECOVERY

Note that for the recovery in Eq. (2), we place either an averaging operation, or linear layer following $W$ with identical row elements. This is because in Eq. (2), the attacker needs $\frac{\partial \mathcal{L}}{\partial b_i} = \frac{\partial \mathcal{L}}{\partial b_{i+1}}$. A sufficient condition for this is that the operation proceeding the imprint module, such as averaging,

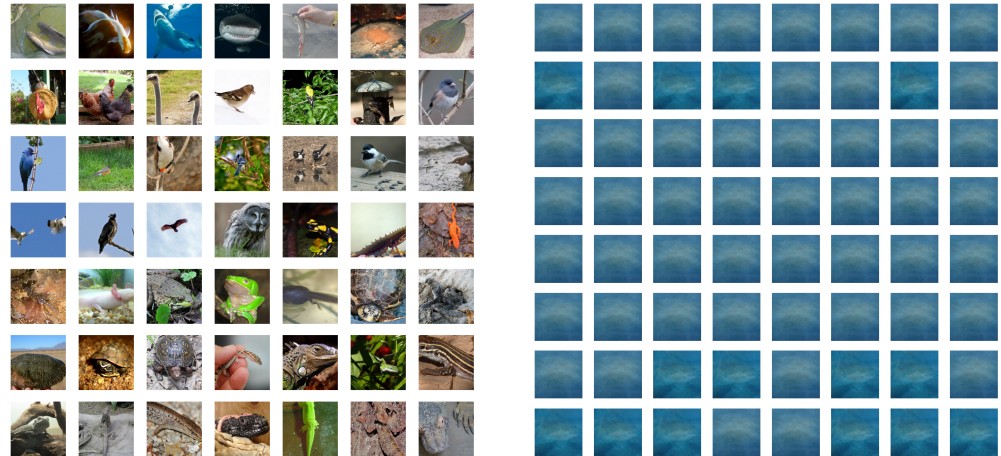

Figure 11: **Left:** Analytic reconstruction for a linear model of 64 ImageNet images with separate classes (PSNR: 36.45 versus true user data). **Right:** Same recovery algorithm but for 64 images from the same class (white shark), (PSNR: 13.84 versus true user data.)

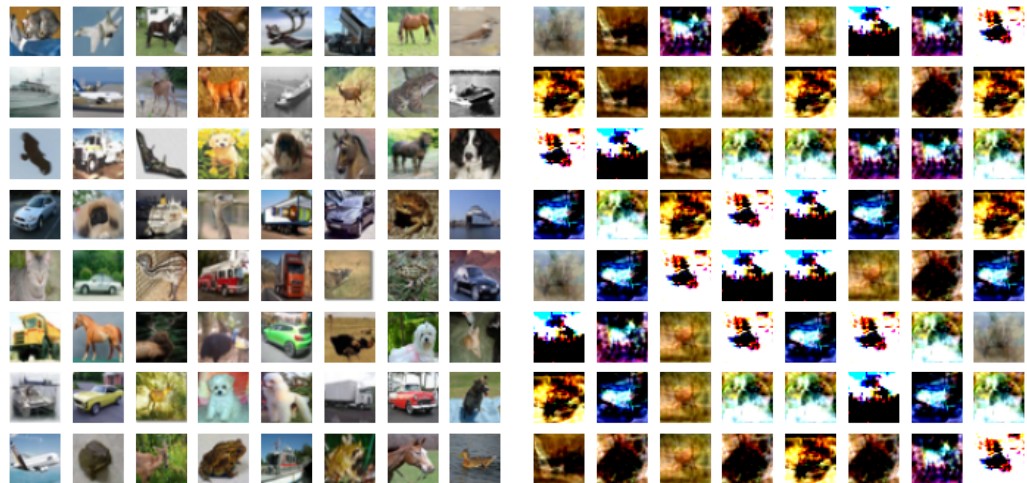

Figure 12: **Left (a)**: A batch of 64 CIFAR10 images. **Right (b)**: The same batch of images reconstructed naively using Eq. (2).

can be expressed as a matrix operation with identical row elements. This falls squarely within our assumed threat model and is empirically how we implement our attack.

## C CODE RELEASE

We provide open-source implementations of all attacks investigated in this work. The repository at `https://github.com/lhfowl/robbing_the_fed` shows a minimalistic implementation of the principles of this attack and the repository at `https://github.com/JonasGeiping/breaching` embeds the attack in a larger framework of privacy attacks against federated learning.

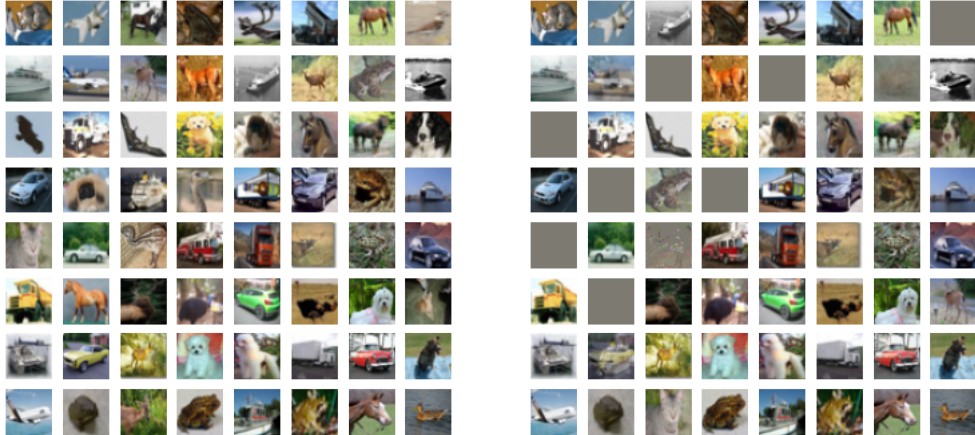

Figure 13: **Left (a)**: A batch of 64 CIFAR10 images. **Right (b)**: The same batch of images reconstructed using the imprint module with 300 bins. Gray images can result from collisions within a given bin.

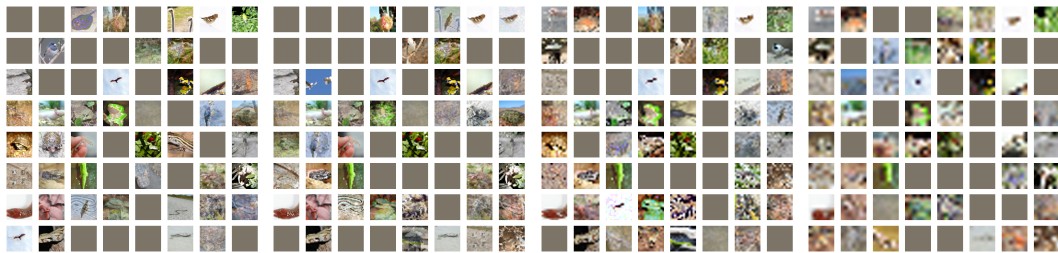

Figure 14: Different placements of the imprint module in a ResNet-18. From left to right: Before the first block ($56x56$), before the third block ($28x28$), before the fourth block ($14x14$) and before the last average pooling ($7x7$). Compare to a placement before the first convolution ($224x224$) and raw input data in Fig. 2. Enlarged versions of each panel can be seen in Figs. 15 - 18

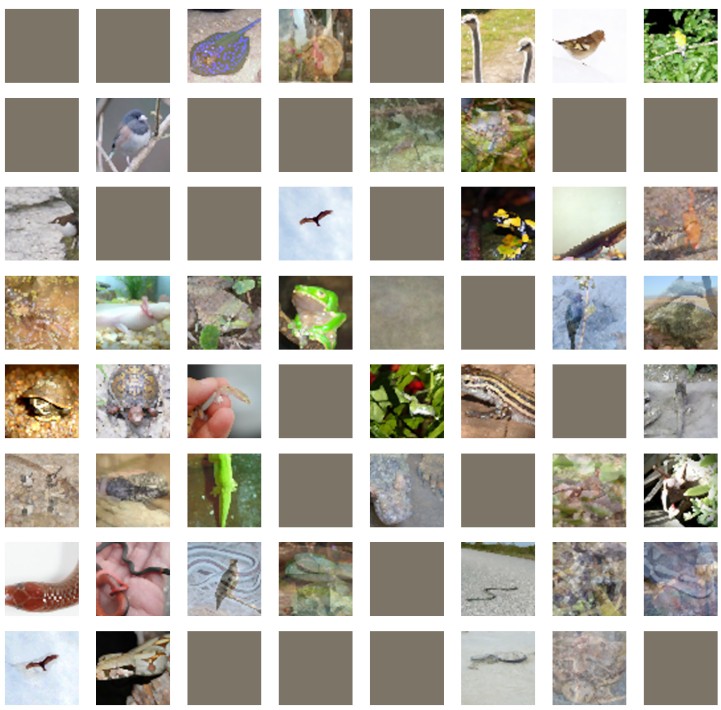

Figure 15: Different placements of the imprint module in a ResNet-18. Before the first block (56 × 56). Only the first three channels at this position are utilized in this demonstration.

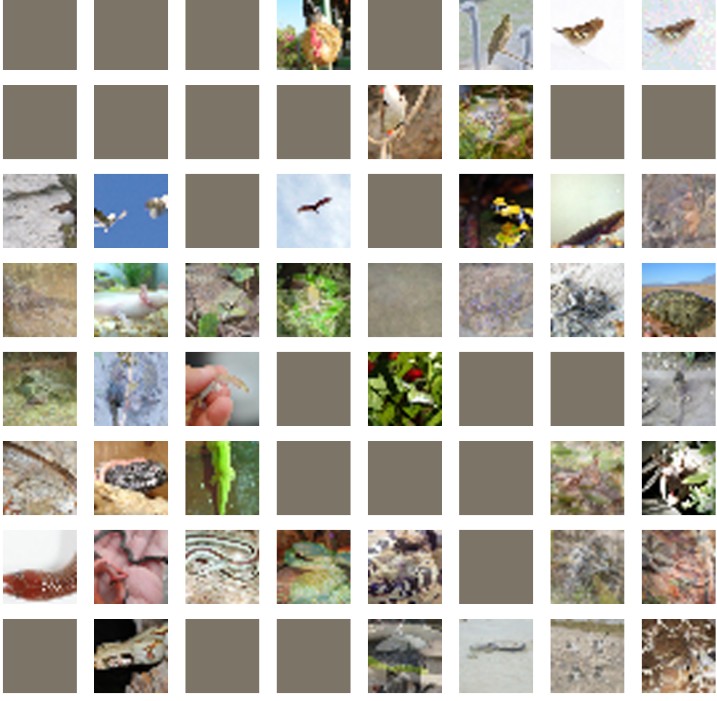

Figure 16: Different placements of the imprint module in a ResNet-18. Before the third block (28 × 28). Only the first three channels at this position are utilized in this demonstration.

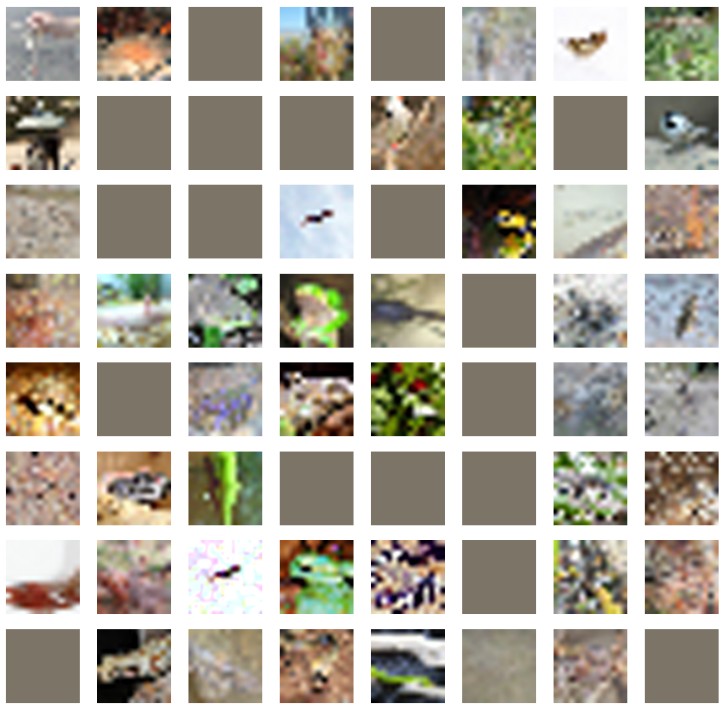

Figure 17: Different placements of the imprint module in a ResNet-18. Before the fourth block $(14 \times 14)$

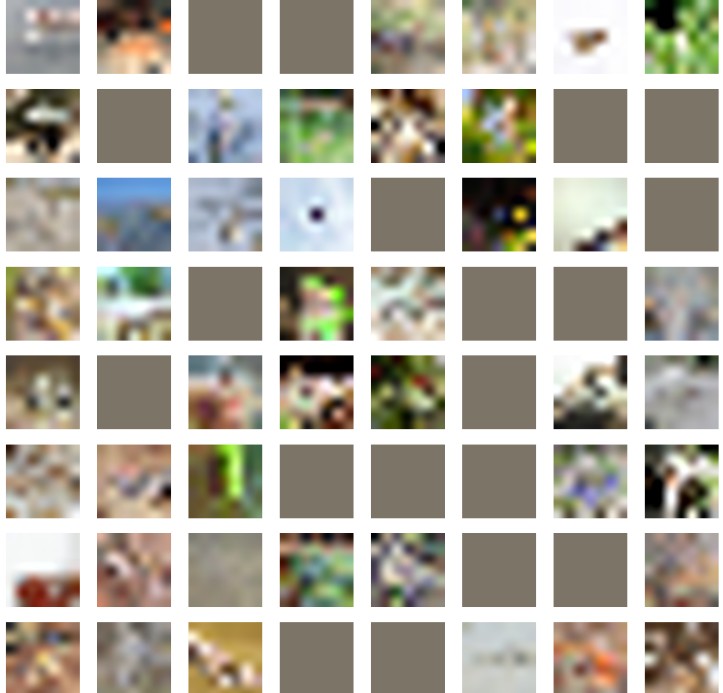

Figure 18: Different placements of the imprint module in a ResNet-18. Before the last average-pooling layer $(7 \times 7)$. Only the first three channels at this position are utilized in this demonstration.

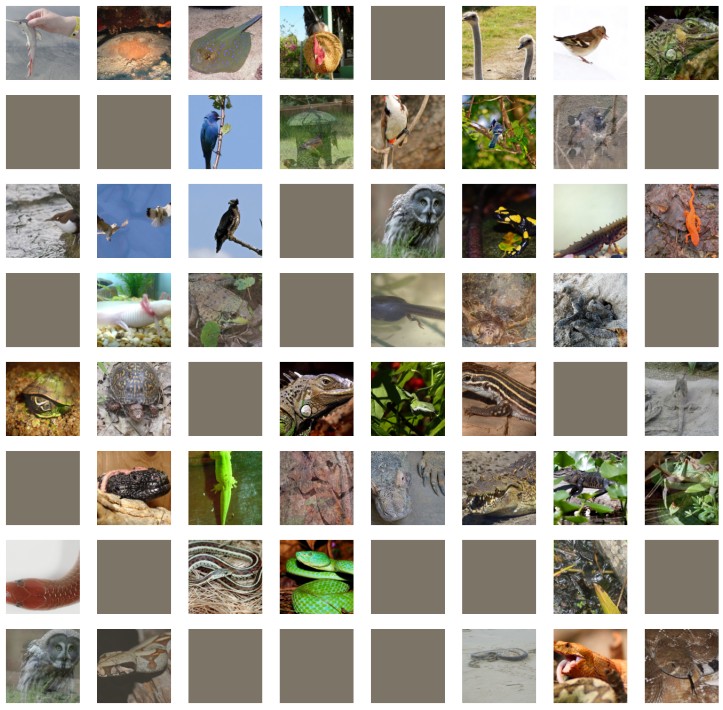

Figure 19: Results for federated averaging for 8 steps with 8 images each, i.e. 64 unique data points for a single user, and 128 bins. PSNR: 32.65. IIP: 70.31%. Drift of bins during local updates leads to a few duplicated entries.

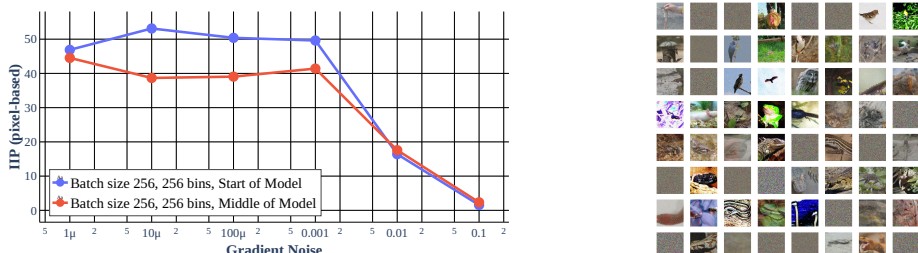

Figure 20: **Left:** IIP score vs Laplacian gradient noise. **Right:** Exemplary recovery for $\sigma = 0.01$. Recovery is stable for a large range of Laplacian gradient noise injections.

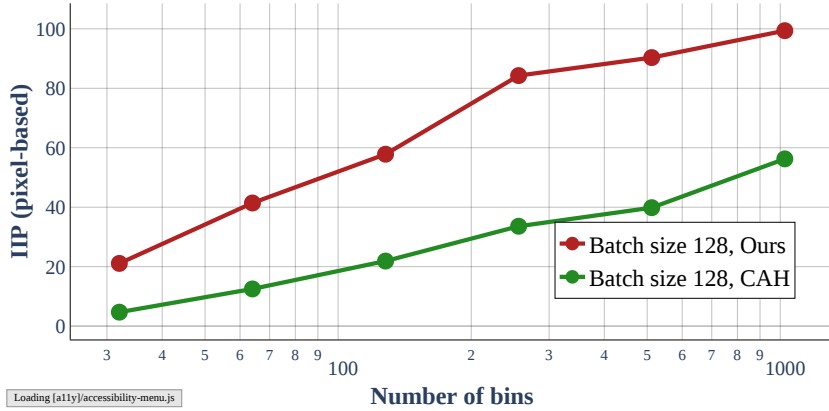

Figure 21: Comparison to the "Curious Abandon Honesty" (CAH) attack proposed in Boenisch et al. (2021). We find our attack outperforms their attack at every scale.

