# OpenReview forum: "Robbing the Fed:  Directly Obtaining Private Data in Federated Learning with Modified Models"
_ICLR.cc/2022/Conference — ICLR 2022 Poster_

### Official Review · Reviewer_AxhH · 2021-10-22

**Correctness:** 4
**Technical Novelty And Significance:** 2
**Empirical Novelty And Significance:** 2
**Recommendation:** 6
**Confidence:** 4

**Main Review:**

Strengths:
- Provides clear and easily understood high level message: Federated Learning is very vulnerable if the server cannot be trusted.
- The paper does not rely on simplifying and unrealistic assumptions. It would be (reasonably) simple to implement the method in practice.

Weaknesses:
- Paper's builds heavily on key insights from previous works, such that from the previous works one could deduce the results in this paper without too much difficulty. And this insight came from previous works on privacy in deep learning, meaning this paper is not making a contribution by newly bringing this insight into the community.

**Summary Of The Paper:**

This paper explores the privacy of federated learning when the server is malicious, specifically that it is allowed to control the architecture and and set certain weights in adversarial ways. By building on a weakness inherent to the RELU activation function, they show that w.h.p the malicious server can obtain exact reproductions of at least some inputs. They then empirically explore how well this works in practice, showing that the method introduced does pose a significant risk to user data privacy.

**Summary Of The Review:**

The paper provides another piece of evidence that federated learning on its own is not sufficient to protect user privacy, and explains how even with only a minor malicious change the server can exactly recover user data. And the message is very clear and well exposed in the paper. However method is only a slight variation on ideas from previous work.

---

> ### Author Response · Authors · 2021-11-18
> **Thank you for your review**
>
> Thank you for your time and thoughtful feedback. Below, we address specific points you raised. Note that the order of some figures/results has changed with the rebuttal revision, and all references in our response are with respect to newer (rebuttal revision) version of our manuscript.
>
> &nbsp;
>
> >Paper's builds heavily on key insights from previous works
>
> While we do explore reconstruction from a linear layer as a toy example, to the best of our knowledge, we are the first attack on federated learning using malicious model modifications. We are also the first attack that can recover images from large batches at high fidelity.   We are not aware of any other publication that uses methods analogous to our imprint module, although there are a range of existing attacks on privacy that recover images using different/unrelated means.
>
> Finally, while the crux of our method - the binning trick - is mechanically simple, we cannot find anything in the literature like this, and we believe that it would be disingenuous for us to present an overly complicated method when a straightforward one will do. Much of the existing literature in the “honest-but-curious” setting has focussed on complex optimization-based attacks with a huge computational budget, but we show that a move to a malicious model with the addition of a few parameters can lead to a much stronger, yet mechanically much simpler attack.  Our proposed attack is applicable in settings that previous attacks cannot handle (such as image extraction from large training batches, other data modalities, etc.).

---

### Official Review · Reviewer_t8bw · 2021-11-02

**Correctness:** 4
**Technical Novelty And Significance:** 4
**Empirical Novelty And Significance:** 4
**Recommendation:** 8
**Confidence:** 4

**Main Review:**

Strengths
1. Well-written and organized paper. The threat models and attacks are clearly described and well-motivated.
2. The proposed imprint module is very novel and exploits the vulnerabilities of gradient computation in popular network design.
3. The attack experiments are well-executed. The reconstructed results are really close to the original inputs, even for gradients aggregated over a large batch size or multiple local iterations or with local DP, which is much better than reconstructing the averaged inputs as most previous works do.

Weaknesses and questions
1. Stealthiness of the attacks, as acknowledged by the authors, might be a problem and the attacks could be easily detected by inspecting the model architecture or parameters. Could the parameters in the imprint modules themselves reveal the attacks, given their distribution might be very different from other layers? Is it possible to disguise the parameters as normal layer parameters as well?
2. Since the imprint module changes the intermediate activations, how does this affect the overall utility of the models? Unfortunately, I did not see any results on how the architecture might change the model performance. If the performance is surprisingly different for a data on device, then the device owner might know there is something wrong with the particular model received from the server.
3. I did not understand the setup for the federated learning experiments in Section 4.3. How many users are sampled in each iteration? How are their data splitted? Does the attack work even when there are multiple local iterations and a large number of users sampled in the central iteration? I.e. aggregation of model updates from a large number of users (i.e. magnitude of thousands) trained with multiple local SGD (which could be the setup when large tech companies deploy FL)
4. There could be another DP related mitigation where the model updates are first securely aggregated and then added with central DP noise before sending to the server. Could the server still infer anything from the noised aggregate?


**Summary Of The Paper:**

This paper introduces novel attacks for reconstructing training data given the device contribution in the federated learning setting. The attacks exploit the gradient behavior of ReLU and insert carefully tailored feed forward layers into a benign network so as to reconstruct the input from the gradients. The demonstrated examples show the effectiveness of the attacks that even training with large batch size, attacks with certain knowledge about the distribution of the data could recover almost the exact training data.


**Summary Of The Review:**

This paper proposed novel and effective data reconstruction attacks in federated learning. Overall, I recommend accepting this paper for its solid technical contribution and convincing results.

---

> ### Author Response · Authors · 2021-11-18
> **Thank you for your review**
>
>
> Thank you for your time and thoughtful feedback. Below, we address specific points you raised. Note that the order of some figures/results has changed with the rebuttal revision, and all references in our response are with respect to newer (rebuttal revision) version of our manuscript.
>
> &nbsp;
>
> > On stealthiness of the attacks
>
> You are correct that in its simplest form, the imprint module could be easy to detect as all the rows would be identically 1/n (in the case of measuring image brightness). We have included a  brief discussion on such tactics in Appendix A.6. However, the imprint module could be adapted to many of the heuristics we could think of. For example, the imprint module could use less conspicuous parameters, like a Gaussian vector, to compute $h(x)$ i.e. $W^i = y$ where $y \sim N(\vec{0}, \sigma \cdot Id)$ instead of brightness. One could then further look at the rank of each layer to detect the imprint module as the imprint module, in its most vanilla construction, always has rank $1$ in its linear layer. However, this is easily evaded by adding a tiny amount of salt-and pepper noise to each row in $W_*$, or mixing in rows that perform “normal” functions in the network. Even performance of the overall model is not a good indicator of the imprint module as the proceeding model can train normally with an imprint module which can be connected by a shortcut connection.
>
> &nbsp;
>
> > Since the imprint module changes the intermediate activations, how does this affect the overall utility of the models?
>
> The inclusion of the imprint module need not affect the performance of the proceeding model. There are several simple ways to ensure this. For example, the server could add a shortcut connection from the input layer, and add the mean of the activations of the imprint module to all the inputs ensuring gradients flow backward through the module but that the impact of the module is a constant addition to intermediate features at this stage.
>
> &nbsp;
>
> > I did not understand the setup for the federated learning experiments in Section 4.3 …
>
> Thank you for raising this confusion. We have since added clarifying sentences to Section 4.3. Are there specific experimental settings you would like explained? In the simplest setting, for a single local update, the only quantity of relevance to the imprint module is the total amount of data. That is to say, 10 users each sending updates on $100$ datapoints (and these updates are securely aggregated) is equivalent (reconstruction-wise) to one user sending an update on $1000$ datapoints. Because of this, for Figure 2, we only specify the number of total images. These $64$ images could be split as $8$ users sending updates on $8$ images, or a single user sending an update on $64$ images.
>
> For multiple local updates, we have clarified for the experiment discussed in Section 4.3 and shown in Figure 20. For simplicity, here we consider 64 images for a single user. We split these equally over their update steps. Each of their 8 local updates is based on 8 different images from this batch of 64 local data points.
>
> In terms of the setting of averaging over thousands of users, each taking multiple local update steps, there is nothing structurally limiting about the number of users in this setting. The server could simply include more bins in the imprint module, or perform a “one-shot” or “many-shot” attack to recover user data.
>
> The same mostly holds for the number of local update steps. The malicious server would have to worry about bin “drift” during repeated local update steps as the biases of the imprint module could change slightly which can weaken the attack, but the server can take mitigating steps (for example minimizing the magnitude of the gradients w.r.t. the biases). Further, the number of local update steps is usually limited, whereas the number of users is potentially unlimited.
>
> &nbsp;
>
> Does this address your questions/concerns?

---

### Official Review · Reviewer_nVzn · 2021-11-02

**Correctness:** 3
**Technical Novelty And Significance:** 3
**Empirical Novelty And Significance:** 3
**Recommendation:** 6
**Confidence:** 2

**Main Review:**

The result (Proposition 1) in the imprint module section has a strong assumption that the server knows the CDF of a quantity associated with user data.  This seems contradictory to the statement in the paper that claims the results here are more data agnostic.

Although the imprint module can be inserted in any position in a neural network (subject to a constraint), I am left wondering whether it would make it better for the attack to put the imprint module earlier or later in the neural network.  I would be interested to see results when testing different positions in the neural network.

The experiments show the power of the attack, while demonstrating that not every image is reconstructed.  Regarding Figure 4, I understand that the license plate is blocked for privacy reasons, but I wonder if the reconstructed image still has a readable license plate.

In the potential defense and mitigation strategies section, I would be interested to know the overall local privacy guarantee, in particular what dimension are the updates?  I would expect sigma = 0.01 to offer better privacy guarantees, but maybe the updates are high dimensional.  I would have liked this section to be more fleshed out.

### UPDATE ###
I have read over the author feedback and think the suggested changes can improve the work.  I will keep my score unchanged.


**Summary Of The Paper:**

This paper presents a new attack on federated learning, demonstrating that model updates shared in a federated setting can still leak user data.  Although federated learning has been used as a privacy technique, this paper shows that other mitigation techniques should also be used.  Attacks before were based on updated from a single data point, and hence aggregating data in an update was used as one approach against this attack.  The adversarial setting here is with a malicious server who may modify the model’s architecture in a seemingly innocuous way.  Allowing models to be changed in ways that are currently allowed with FL APIs, this paper shows that user data can still be reconstructed even with large aggregations.  Another benefit of this approach is that it is more data agnostic, not relying on strong data priors.


**Summary Of The Review:**

Overall a nice attack that shows the keeping data distributed while only sharing model updates does not provide strong privacy guarantees.

---

> ### Author Response · Authors · 2021-11-18
> **Thank you for your feedback**
>
> Thank you for your time and thoughtful feedback. Below, we address specific points you raised. Note that the order of some figures/results has changed with the rebuttal revision, and all references in our response are with respect to newer (rebuttal revision) version of our manuscript.
>
> &nbsp;
>
> > The result (Proposition 1) in the imprint module section has a strong assumption…
>
> Thank you for bringing this to our attention. We would like to clarify that the attacker does not need to know the full distribution of the data, but only the CDF of some scalar value that can be computed from data by a linear function (e.g. the distribution of image brightness).
>
> Also, to clarify, the assumptions of Prop. 1 are not necessary for a successful attack, only for the guarantee on expected data recovered. In fact, in *all* the experiments in the paper, we do not assume knowledge of the CDF of the quantity we measure. Instead, we only assume to know the mean and variance of this quantity - a realistic assumption for average brightness of most image data.  We have since included a comparison of the true distribution of brightness to our estimated distribution in Appendix A2. In practice, an inexact CDF will work.
>
> As to the practicality of the assumption itself, we believe this is a fairly weak assumption that is easily fulfilled in practice. The FL system is not trained in a vacuum, the server has to have a rough idea what kind of data will be used (for example to choose a good architecture). Because our attack is based only on a scalar quantity, a few generic samples could be used to estimate it. For example, average pixel brightness is very likely a good choice for any type of image data, and an organization likely has some idea what brightness distribution to expect in its training images. We have included a plot of the estimated density function for brightness of ImageNet images using different proportions of the dataset to estimate this quantity. We find that even with 0.1% of the dataset, the server could get a good estimate of the distribution of brightness (see Appendix Fig. 6).
>
> Finally, as to our statement about being data-agnostic, this is in reference to the data type. Previous optimization based methods rely heavily on strong regularizers that exist in the image domain, but not in other domains of interest, like text. We do not rely on any such assumptions on the type of data.
>
> &nbsp;
>
> > I am left wondering whether it would make it better for the attack to put the imprint module earlier or later in the neural network ...
>
> You are correct that the placement of the imprint module can affect results, especially in the image domain. We study this in Figure 4, as well as Appendix Figures 15-19. Placing the imprint module later in the network has the benefit of requiring fewer parameters, but on the flip side only a downsampled image can be recovered for the ResNet architecture where we investigate this. For convolutional architectures like the ResNet, the placement of the imprint block is a trade-off between conspicuousness of the attack and resolution of the recovered data. For other network architectures without downsampling stages, the block can be placed in any position in the network with the same effect.
>
> &nbsp;
>
> > I understand that the license plate is blocked for privacy reasons, but I wonder if the reconstructed image still has a readable license plate.
>
> Yes, the license plate is indeed visible and legible in the non-redacted recovered image. Note how the smaller “Ford” logo is still readable in the reconstruction. We computed the PSNR of the recovered data (161.36) based on the unblocked image. A PSNR of this magnitude is at floating point machine precision, meaning that every bit of the original 0-255 image representation is recovered perfectly, and a human cannot discern between the original and recovered image. In fact, PSNR>100 corresponds to less than 1 bit flip for an ImageNet sized image with a bit depth of 16. To comply with ICLR ethical guidelines we have blocked the license plate for publication on openreview.
>
> &nbsp;
>
> > I would be interested to know the overall local privacy guarantee, in particular what dimension are the updates? …
>
> For Figure 5, these updates are coming from the imprint module before a ResNet-18 which collectively contain approximately 49M parameters. From a privacy standpoint however, all of the parameters belonging to the original ResNet-18 network are not used during the attack and can be maliciously modified to have arbitrarily small gradient contributions. To recover input data in full resolution, the dimension of the update of the imprint block gradients is approximately equal to the dimension of the input data (~150k for ImageNet) times the number of bins.
>
> &nbsp;
>
> Does this address your questions/concerns?

---

### Official Review · Reviewer_Pq3S · 2021-11-04

**Correctness:** 3
**Technical Novelty And Significance:** 2
**Empirical Novelty And Significance:** 2
**Recommendation:** 5
**Confidence:** 2

**Main Review:**

The idea looks interesting, but I have some concerns as below.

1. The assumption used in Proposition 1 is not realistic
    1. I’m not sure how come the server knows the CDF of “some quantity associated with user data”. By the way, would you clarify “some quantity associated with user data”? It is vague.
2. Does the reconstruction available for all cases?
    1. It seems like several figures has gray boxes, meaning that the image is not recovered. Do we have some understanding why/when such failure cases occur? If not, it is better to delve into what happens to that failure scenarios and what is the possible fix for that failure.
3. No comparison with related works?
    1. I have no idea what are other schemes handling malicious server scenario, but if we can find some, it is necessary to compare this scheme with conventional schemes.


**Summary Of The Paper:**

This paper provides a privacy attack scheme for federated learning, retrieving the private data from the model aggregation techniques.


**Summary Of The Review:**

This is an interesting paper, but it is better to include more concrete theoretic/empirical analysis on why/when the suggested scheme works. Moreover, comparison with existing schemes is missing.

---

> ### Author Response · Authors · 2021-11-18
> **Thank you for your review**
>
> Thank you for your time and thoughtful feedback. Below, we address specific points you raised. Note that the order of some figures/results has changed with the rebuttal revision, and all references in our response are with respect to newer version of our manuscript.
>
> &nbsp;
>
> > The assumption used in Proposition 1 is not realistic…
>
> In case there was ambiguity in our presentation of the proposition, we would like to stress that the attacker does not need to know the full distribution of the data, but only the CDF of some scalar value that can be computed from data by a linear function (e.g. the distribution of image brightness).
>
> Also, to clarify, the assumptions of Prop. 1 are not necessary for a successful attack, only for the guarantee on expected data recovered. In fact, in *all* the experiments in the paper, we do not assume knowledge of the CDF of the quantity we measure. Instead, we only assume the mean and variance of this quantity - a realistic assumption for average brightness of most image data.  We have since included a comparison of the true distribution of brightness to our estimated distribution in Appendix A2. In practice, an inexact CDF will work.
>
> As to the practicality of the assumption itself, we believe this is a fairly weak assumption that is easily fulfilled in practice. The FL system is not trained in a vacuum, the server has to have a rough idea what kind of data will be used (for example to choose a good architecture). Because our attack is based only on a scalar quantity, a few generic samples could be used to estimate it. For example, average pixel brightness is very likely a good choice for any type of image data, and an organization likely has some idea what brightness distribution to expect in its training images. We have included a plot of the estimated density function for brightness of ImageNet images using different proportions of the dataset to estimate this quantity. We find that even with 0.1% of the dataset, the server could get a good estimate of the distribution of brightness (see Appendix Fig. 6).
>
> &nbsp;
>
> > would you clarify “some quantity associated with user data”?
>
> Thank you for pointing out this ambiguity. We have since further clarified this in section 3.3 of the updated manuscript. In general, a quantity is any linear function $h:\mathbb{R}^n \rightarrow \mathbb{R}$ of the user data about which the server knows the CDF. In our experiments, we choose image brightness as the quantity of interest. The network weights are set up so that images with different levels of brightness (the “quantity” of interest in this case) activate different neurons so that these images can be unmixed during the recovery process.
>
> &nbsp;
>
>
> > It seems like several figures has gray boxes, meaning that the image is not recovered...
>
> The reason why a box is gray is that no data from the sampled batch fell into the corresponding “bin” of the CDF. For example, if no image has brightness corresponding to bin $i$, then $\nabla_{W_*^{l}}\mathcal{L} - \nabla_{W_*^{l+1}} \mathcal{L} = \vec{0}$ and the "image" recovered is simply gray.  In the caption for Fig. 2, we briefly describe this behavior, and we theoretically analyze this behavior as a consequence of Prop. 1 in the proof. In Fig. 1a we plot the expected percentage of recovered data points as a function of number of bins.
>
> &nbsp;
>
> > I have no idea what are other schemes handling malicious server scenario…
>
> To the best of our knowledge, we are the first attack on privacy that explicitly modifies the architecture/parameters of a FL model. Because of this, our method is not directly comparable to existing methods.
>
> Nonetheless, the malicious server model that we consider in this work is closely related to the more commonly analyzed honest-but-curious server model (see Section 2), but leads to an astronomically stronger recovery.
>
> For a direct comparison, we have now added results for an optimization-based attack against an honest server for the scenario of Fig. 2 (batch size 64, ResNet 18) to the appendix (Fig. 9). Attacks in the honest scenario have trouble recovering most data even for a batch size of 8 (which is very small in more realistic FL settings). This can also be seen by comparing directly with the results of Yin et al. 2021 (cf. our references), who reconstruct up to a PSNR of 12.93 for a batch of size 8 (here we take numbers from their paper as code has not been released for this project.)  For comparison, our proposed attack reaches a PSNR of 82.20 on a much larger batch size (64, with 128 bins), and could be arbitrarily stronger with additional bins.  In summary, our proposed method recovers user images with much higher accuracy than other attacks that have been demonstrated on federated learning, however it also requires a different threat model (modifying network parameters) so it is not appropriate to say that our attack is strictly “stronger”.
>
> &nbsp;
>
> Does this address your questions/concerns?

---

### Public Comment · ~Jingwei_Sun2 · 2021-11-09
**Related work**

It is a nice work! Just a kind remind, the vulnerability analysis seems similar to [1]. Maybe you want to briefly discuss in the paper whether your attack can be effective when the clients apply the defense in [1].

[1] Sun, J., Li, A., Wang, B., Yang, H., Li, H., & Chen, Y. (2021). Soteria: Provable Defense Against Privacy Leakage in Federated Learning From Representation Perspective. In Proceedings of the IEEE/CVF Conference on Computer Vision and Pattern Recognition (pp. 9311-9319).

---

> ### Author Response · Authors · 2021-11-18
> **Thank you for your comment.**
>
> Thank you for pointing us to this work, we have added a reference to the vulnerability of Sun et al. 2021 to our submission. The defense is certainly interesting. An adaptive attack against it (based on skipping the defended layer) has been investigated in Balunović et al 2021, “Bayesian Framework for Gradient Leakage” for the honest-but-curious threat model. For the malicious server threat model we consider in this work, we note that control over the parameters of the model gives the attacker a lot of room to circumvent the perturbed representation, as the attacker chooses the function f. Without such modifications we believe that the defense is further covered by our analysis of gradient noise in Fig. 5 in the case where the imprint module is included in the beginning of a network.

---

### Author Response · Authors · 2021-11-29
**General Summary of Author Response to Reviewers**

We thank all the reviewers for their constructive feedback on our work. Below, we summarize the main constructive reviews that we received and what we have done to address these concerns. We have included more detailed discussion in the individual responses.

1. Reviewers **Pq3S**, **nVzn** raised concerns about the assumptions of Proposition 1 in our manuscript. Specifically, concerns were raised about how realistic it is for the attacker to know information about the quantity, $h$, associated with the user data.

    - **What we have done**: We have since clarified ambiguity about the assumptions of Proposition 1 in our updated manuscript. We think these assumptions are not as strong as they might have seemed. We have updated our manuscript with several figures investigating different choices of $h$, as well as approximating distributions of $h$. We also stress that all the empirical results we present in the paper do *not* assume perfect knowledge of the underlying distribution of $h$, but only the mean and variance of $h$ - a realistic assumption in many cases.


2. Reviewers **nVzn**, **t8bw** asked about secure aggregation, multiple local updates, and DP techniques, and their effects on our proposed attack.
    - **What we have done**: We have since added some more explanation to sections 4.3, 5 in our updated manuscript. To summarize, in the simplest case, our attack does not depend on the number of users in an aggregated update, only the *total* amount of data. So 100 users sending updates on 10 images is the same (for our attack) as 10 users sending updates on 100 images. When multiple local updates are used, the attacker could appropriately adjust parameters to all but eliminate any degradation in the attack success (see section 4.3). Finally, in terms of DP techniques, in the simplest case, when an update is clipped, and noise is added, for images this just corresponds one-to-one with adding scaled noise to the input data. We study this setting in Figure 5.


We appreciate the time the reviewers have dedicated to our work, and we are eager to hear if our responses have alleviated the initial concerns raised.

---

### Decision · Program_Chairs · 2022-01-20

**Decision:**

Accept (Poster)

**Comment:**

The authors provide an interesting improvement on privacy attacks in federated learning, demonstrating the ability to extract individual points even over large batches. While there were some concerns about the technical difficulty of the approach, reviewers were broadly in support of the work. As I tend to agree, this is an interesting strengthening beyond what it appears we were able to do before. This is yet another piece of evidence against the canard in FL that only sharing gradient updates provides privacy guarantees.